# Larger is Better in the Parasitoid *Eretmocerus warrae* (Hymenoptera: Aphelinidae)

**DOI:** 10.3390/insects11010039

**Published:** 2020-01-03

**Authors:** Tao Wang, Michael A. Keller

**Affiliations:** 1Guangdong Public Laboratory of Wild Animal Conservation and Utilization, Guangdong Key Laboratory of Animal Conservation and Resource Utilization, Guangdong Institute of Applied Biological Resources, Guangzhou 510260, China; 2School of Agriculture, Food and Wine, Waite Campus, University of Adelaide, Adelaide 5005, Australia; mike.keller@adelaide.edu.au

**Keywords:** body-size, daily oviposition, longevity, parasitoid, *Eretmocerus warrae*, *Trialeurodes vaporariorum*, immature developmental temperature

## Abstract

*Eretmocerus warrae* (Hymenoptera: Aphelinidae) is a specialist parasitoid that is used for the control of the greenhouse whitefly, *Trialeurodes vaporariorum* (Hemiptera: Aleyrodidae). We investigated how temperature affects the body-size, life-time oviposition, and longevity of *E. warrae* at different stages of life. The body-sizes of both this parasitoid and its host are influenced by temperature. Body-volume indices that reflect body-sizes fell by 47.7 % in *T. vaporariorum* compared with 57.6% in *E. warrae* when temperature increased from 20 to 32 °C. The life-time oviposition of female adults of *E. warrae* that grew at the immature developmental temperature of 20 °C was 86 ± 22 eggs, more than 66 ± 11 eggs at 26 °C, and 65 ± 23 eggs at 32 °C. Besides the influence on fecundity, temperature also influences the oviposition behaviour at the adult stage. More eggs were oviposited at 20 and 26 °C than at 32 °C. Higher temperatures reduced survival in the immature developmental stages and longevity in adults. Adult females lived for a maximum of 8.9 ± 1.8 days at 20 °C and laid a maximum of 97.4 ± 23.2 eggs when reared at 20 °C and maintained at 26 °C as adults. Adult body-size is positively correlated with life-time oviposition but not adult longevity. The results imply that temperature influences the nature of interactions between a parasitoid and its host. Larger wasps can live longer and parasitise more hosts, which should improve their performance as biological control agents.

## 1. Introduction

Temperature is an ecological factor that moderates many aspects of insect fitness, including body-size, longevity, and oviposition. Insects are poikilothermic, so temperature influences physiology and development. Immature development in relatively cooler conditions often leads to a larger body-size [1,2,3]. This change in body-size affects fecundity in a more complicated manner. Angilletta et al. [2] found that individuals grow slower in colder ambient environments, and argued that there should be no substantial increase in fecundity in such conditions. However, adult insects are usually more active in warmer conditions, so higher temperatures generally lead to a higher oviposition rate [4]. Warmer environmental conditions result in higher levels of metabolism and higher developmental rates. Thus, the longevity of insects normally has an inverse relationship with the ambient temperature [3,5]. Therefore, temperature can affect the number of eggs that are laid during an individual’s lifetime.

The body-size, fecundity and longevity of insects are interrelated. Generally, a larger body-size means more ovarioles and a higher fecundity [6]. Being larger can also result in greater longevity in some species [7], but not others [8]. Moreover, there are potential trade-offs between fecundity and longevity. Higher fecundity might result in a higher oviposition rate and cause higher energy consumption, and thus reduce longevity. Longer lifespans in insects give them more chances to oviposit and usually lead to higher life-time oviposition [9].

Temperature affects parasitoids directly and indirectly through the responses of host insects and plants. Plants can display physical differences and chemical changes in response to temperature [10]. These changing plant characteristics may directly influence the development and behaviour of host insects, and thus potentially influence the parasitoids directly or indirectly [11]. The size and quality of host insects will also vary under different temperature conditions. Hosts that are larger or offer better quality resources to parasitoids generally deliver better fitness gains to parasitoids that develop on them [12]. Host size is particularly key to solitary species of parasitoids since parasitoids that emerge from larger hosts tend to be larger. However, the level of host acceptance might be higher since there might be higher levels of defence in hosts of higher fitness gain.

The research reported here investigated how temperature affects the body-size, life-time oviposition, and longevity of an idiobiont parasitoid and its host. Of particular interest is oviposition over the life-time of parasitoids, which is directly related to the life-time parasitism of hosts and therefore contributes to the level of biological control of host insects. It is also valuable to understand how temperature affects the biology of parasitoids, to assist with developing optimal rearing procedures for commercial production.

The model parasitoid used in this research was *Eretmocerus warrae* (Hymenoptera: Aphelinidae). Its host insect is the greenhouse whitefly, *Trialeurodes vaporariorum* (Hemiptera: Aleyrodidae). The greenhouse whitefly attacks a wide variety of agricultural and horticultural plants from 249 genera and 84 angiosperm plant families [13]. Tomato, *Solanum lycopersicum*, was the host plant used in this investigation. *E. warrae* has been developed as a commercial product by Biological Services (Loxton, South Australia). There has been limited research on *E. warrae* and its role in the biological control of whiteflies. Notably, there has been no investigation of the influence of temperature on its body-size, longevity, and life-time oviposition. This research will help with understanding the effects of temperature on aspects of the fitness of *E. warrae* and its host. It will also be useful in the rearing and release of this parasitoid in commercial greenhouses.

## 2. Materials and Methods

A culture of greenhouse whitefly was established from insects that were collected from eggplant, *Solanum melongena*, in greenhouses at the Waite Campus of The University of Adelaide [14]. The whitefly culture was maintained on tobacco, *Nicotiana tabacum*, at 26 °C and a 14 L:10 D photoperiod.

A breeding culture of *E. warrae* was initiated with pupae provided by Biological Services (Loxton, South Australia). It was maintained at 26 °C. When necessary, the pupae of *E. warrae* were kept in an incubator at 8 °C to arrest development. When parasitoids were needed for experiments, pupae were moved from the 8 °C incubator at 20:00 to another incubator, which was set at 26 °C to stimulate them to emerge. Most *E. warrae* emerged the following morning. Honey drops were provided as food to adults.

*E. warrae* prefers to parasitise the second instars of the greenhouse whitefly [15]. To obtain cohorts of second instars, tomato plants (cv. ‘Improved Appolo’) were exposed to ovipositing adult whiteflies for six hours. Then, the adults were blown away with a hairdryer. These plants were grown at selected experimental temperatures until the whiteflies reached the second instar stage. Tomato plants around 50 cm tall with five fully expanded leaves were used in experiments.

Clip cages were used to confine insects on tomato leaves [16]. They were made of two rings of 12 mm thick polyethylene foam that had inside and outside dimensions of 40 mm and 55 mm, respectively, which were held together over a leaf with wire staples pushed into the edges. There was a transparent cellulose acetate sheet on the bottom of each cage, which allowed wasps to be observed, and fine organza on top for aeration. An aspirator made of plastic tubing was used to handle wasps. Honey drops were placed on the organza of clip cages as food for *E. warrae*.

A two-stage experiment was conducted to determine the effects of temperature on both the immature and adult stages of *E. warrae* (Figure 1). The experiment was conducted using three constant temperatures in the range where development typically occurs [14], 20, 26, and 32 °C. Experiments were conducted in three incubators (Adelab Scientific, Thebarton, South Australia, Model 1390D) that were calibrated to means within 0.1 °C of set temperatures with a precision thermometer, (E-MIL, H.J. Elliott Ltd, Treforest, UK) and had a measured variation of ±0.3 °C. The rearing temperatures were set according to the experiments, and the photoperiod was 14 L:10 D.

To characterise the effects of temperature on the growth of the immature greenhouse whitefly, second instars, which had not been exposed to parasitoids, were chosen randomly for body-size measurements. The length and width of each nymph were measured using an ocular micrometer on a dissecting microscope at a magnification of 80x. An index of body-volume was calculated as (length x width) ^3/2^. The body-volume index should be proportional to the actual body-volume. Variations in body-size measurements of greenhouse whitefly among temperatures were subjected to regression analyses with linear and quadratic terms using R version 3.4.0 (21 April 2017) [17]. Parameter estimates are given as mean ± standard deviation since the aim was to estimate life history characteristics.

The effects of temperature on the body-size, fecundity, and longevity of adult *E. warrae* were evaluated as follows. Second instar hosts were exposed to adult *E. warrae* for six hours in clip cages. The female wasp numbers in each clip cage were 4, 2, and 2 at 20, 26, and 32 °C, respectively, since the rate of oviposition was lower at 20 °C. A new replicate was initiated once each week, for five weeks. 

Five adult wasps were chosen randomly from each clip cage when they emerged and placed in individual clip cages at 20, 26, and 32 °C. One tomato leaf infested with second instar greenhouse whitefly nymphs that were reared at 26 °C was provided each day until the wasp died. After exposure to wasps, whitefly nymphs were turned over using an insect-mounting pin to count the number of eggs laid each day. Upon death, the head width and hind-tibia length of each wasp was measured using an ocular micrometer as previously described, and a body-volume index was calculated. The longevity of these wasps was also recorded.

The effects of temperature during the developmental (“immature temperature”) and adult stages (“adult temperature”) on adult longevity and lifetime oviposition were analysed first using linear mixed-effects models (lmer function in package lme4) [18] using R version 3.4.0 (21 April 2017) [18]. The means of the longevity and lifetime oviposition of each of the five females per replicate served as the dependent variables, and replication was treated as a random effect. For adult longevity, both immature and adult temperatures were treated as interval scales. Data were analysed using regression analysis on linear and interaction terms. For lifetime oviposition, treating the adult temperature as an interval scale did not account for the non-linear nature of the biological response, even when a quadratic term was added to the model. Therefore, adult temperatures were considered binary variables, and both simple and interaction terms for the 26 and 32 °C treatments were included in the initial model (20 °C served as a reference treatment in the analysis). Non-significant terms were eliminated from the final analysis. After the forms of the responses to immature and adult temperature were determined, the analyses were repeated on the entire data set with the adult body-volume index of *E. warrae* included to determine its effects. Body-volume indices were multiplied by 1000 so that they were the same magnitude as the temperature in the analyses. The aim of these statistical analyses was to identify the nature of the effects of the independent variables on longevity and lifetime oviposition, rather than to estimate predictive functions. Apparent statistical outliers were identified from large residual values. These outliers were removed, and the data set was reanalysed to determine if there was a pronounced change in the estimated regression coefficients that would indicate a different model structure.

## 3. Results

The body-sizes of greenhouse whitefly nymphs and adult *E. warrae* were affected by temperature. The observed body lengths and widths of the greenhouse whiteflies were smallest at 32 °C (Figure 2a). The mean body length and width of greenhouse whiteflies that developed at the immature developmental temperature of 20 °C were 20.1% and 25.8% longer and wider, respectively, than those that developed at 32 °C. Regression analysis indicated statistically significant (body width of greenhouse whiteflies: F_2,57_ = 42.01, *p* < 0.001; body length: F_2,57_ = 79, *p* < 0.001; body-volume index: F_2,57_ = 62.04, *p* < 0.001) linear and quadratic responses in body-sizes to varying temperature, which indicates overall nonlinear responses to temperature.

Likewise, observed head widths and hind-tibia lengths of *E. warrae* were smallest at 32 °C (Figure 2b). The mean head width and hind-tibia length of *E. warrae* that developed at immature developmental temperature of 20 °C were 57.1% and 54.6% larger, respectively, than those that developed at 32 °C. Body-volume indices derived from linear body measurements also varied in a nonlinear manner (head width of *E. warrae*: F_2,12_ = 6181, *p* < 0.001; hind-tibia length: F_2,12_ = 2068, *p* < 0.001; body-volume index: F_2,12_ = 2068, *p* < 0.001). The greatest decline in the body-size of greenhouse whiteflies occurred between 26 and 32 °C, whereas the sharpest decline for *E. warrae* occurred between 20 and 26 °C (Figure 2). 

When body-size was not included in the analyses, the effects of temperature on longevity and lifetime oviposition differed in their form. There was an interaction between developmental temperature and adult temperature on adult longevity that was fitted well by a linear model with an interaction term between the independent variables (Figure 3a). Longevity declined markedly with increasing developmental temperature when adults were held at 20 °C, whereas there was no clear effect of developmental temperature when adults were held at 32 °C. By contrast, the effects of developmental and adult temperatures on lifetime oviposition could not be fitted with a simple linear model. When adult temperatures were considered binary variables, there were significant effects of the immature temperature, the adult temperature at 32 °C, and the interaction of the adult temperature at 32 °C with the immature temperature on lifetime oviposition (Figure 3b). The number of eggs laid declined sharply as the developmental temperature increased from 20 to 32 °C, and this effect did not differ when adults were held at 20 or 26 °C. However, the numbers of eggs laid by females were reduced when adults were held at 32 °C, but the decline in oviposition was less pronounced than at the lower adult temperatures.

Daily oviposition varied in a consistent manner in response to temperature (F_5,39_ = 165.9, *p* < 0.001). The greatest numbers of eggs were laid each day by wasps that developed at the lowest temperature, and numbers declined as wasps aged (Figure 4). Adult wasps that were held at 26 °C laid the most eggs on the first day, followed by those held at 20 °C, then 32 °C. By the third day of adult life, wasps held at 20 °C laid the largest number of eggs each day followed by 26 °C, then 32 °C.

Lifetime oviposition (Figure 5) and adult longevity (Figure 6) varied with body-volume index both within and among temperature treatments. Within a temperature treatment, larger wasps lived longer and laid more eggs. The fitted regression model indicated that for wasps of any given size, adult longevity was the same at adult temperatures of 20 and 26 °C and decreased as the adult temperature increased from 20 to 32 °C (Figure 6). However, the magnitude of this decrease declined as the immature temperature increased. Longevity increased as the body-volume index increased for each combination of immature and adult temperatures (Figure 6). The fitted lines for longevity vs. body-volume index were collinear for wasps that were reared at 20, 26, and 32 °C and held at the adult temperature of 20 °C. The fitted longevity lines dropped significantly as adult temperatures increased, and this drop was less pronounced for higher temperatures. Removal of outliers did not affect the significant statistical relationships that were identified.

## 4. Discussion

In the present study, temperature was shown to affect the biology of adult *E. warrae* in a number of ways. A cooler temperature during immature development was shown to be associated with larger body-size in both the host and the adult wasp (Figure 2). These changes in size were not parallel. On the one hand, this suggests that the host and the parasitoid have different physiological responses to temperature that mediate their growth and development. On the other hand, temperature may play a more complex role in mediating host plant and host insect responses that affect the parasitoid’s biology [11]. These are not mutually exclusive explanations, and both warrant further investigation. 

A cooler temperature during development produced adult wasps that exhibited greater levels of lifetime oviposition (Figure 3). This was mediated by the temperatures that adults experienced, with lower levels of oviposition at the highest temperature, 32 °C, which is just above the predicted optimum temperature for development of 31.5 °C [15]. There seems to be a complex interaction between body-size, behaviour as revealed by daily patterns of oviposition, and adult longevity (Figure 4). In particular, wasps that were held at 26 °C displayed the highest levels of oviposition on the first day after emergence, but later it was wasps that were held at 20 °C that laid more eggs each day. This interaction between adult temperature and daily oviposition suggests an underlying physiological complexity, as was found in *Aphidius rhopalosiphi* [3].

Adult longevity was shorter as temperature increased from 20 to 32 °C (Figure 3). This is consistent with observations of other parasitoids, such as *Encarsia formosa* [19] and *Cirrospilus* sp. nr. *lyncus* [20].

In adult parasitoids, both lifetime oviposition and adult longevity are the result of intrinsic characteristics that are the outcome of immature development and the biological summation of ageing and activities (e.g., feeding), which is analogous to mathematical integration. Lifetime oviposition and adult longevity were greater in larger individuals that were exposed to the same immature and adult temperature regimes (Figure 5 and Figure 6), which indicates that larger body-size gives a fitness advantage when other factors are equal. A similar finding was recorded for *Anagyrus kmali* [21]. However, this result is inconsistent with patterns documented in an earlier review [8], which did not take into account the environment experienced by immature and adult parasitoids. Further experiments with other species are warranted to determine if this is a more general relationship.

Our findings suggest that the rearing temperature and the resulting effects on body-sizes and performance should be considered when formulating strategies for using parasitoids in biological control in greenhouses. Ideally, biological control agents should be reared in a manner that optimises their capacity to control the target pest. On the one hand, our earlier results showed how controlling the temperature can affect the production of *E. warrae*. As the temperature is raised, the developmental time decreases, which makes the production of cohorts of adults faster [13]. On the other hand, wasps that develop faster are smaller, which causes them to have lower fecundity. Smaller adults are less likely to parasitise larger whitefly nymphs [14]. Controlling the temperature involves costs, so the trade-offs between cost and the efficiency of production and adult wasp quality warrant further investigation.

## 5. Conclusions

Our results give insights into how the development, longevity, and reproduction of *E. warrae* vary in response to temperature. The temperature during development plays a key role in determining the adult body-size, which determines in large part the reproductive potential and longevity of adults. The temperature experienced by adults mediates this potential to ultimately determine the characteristics and fitness of adults. The results provide further insights into the way that rearing conditions can affect the performance of *E. warrae* when it is used in augmentative biological control of the greenhouse whitefly.

## Figures and Tables

**Figure 1 insects-11-00039-f001:**
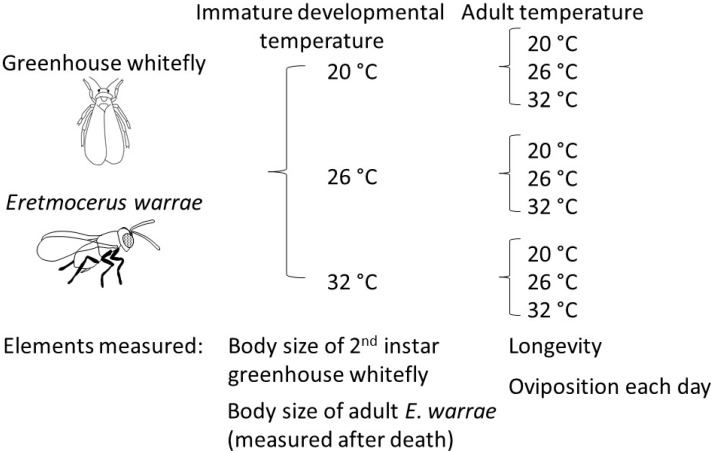
Summary of experimental design.

**Figure 2 insects-11-00039-f002:**
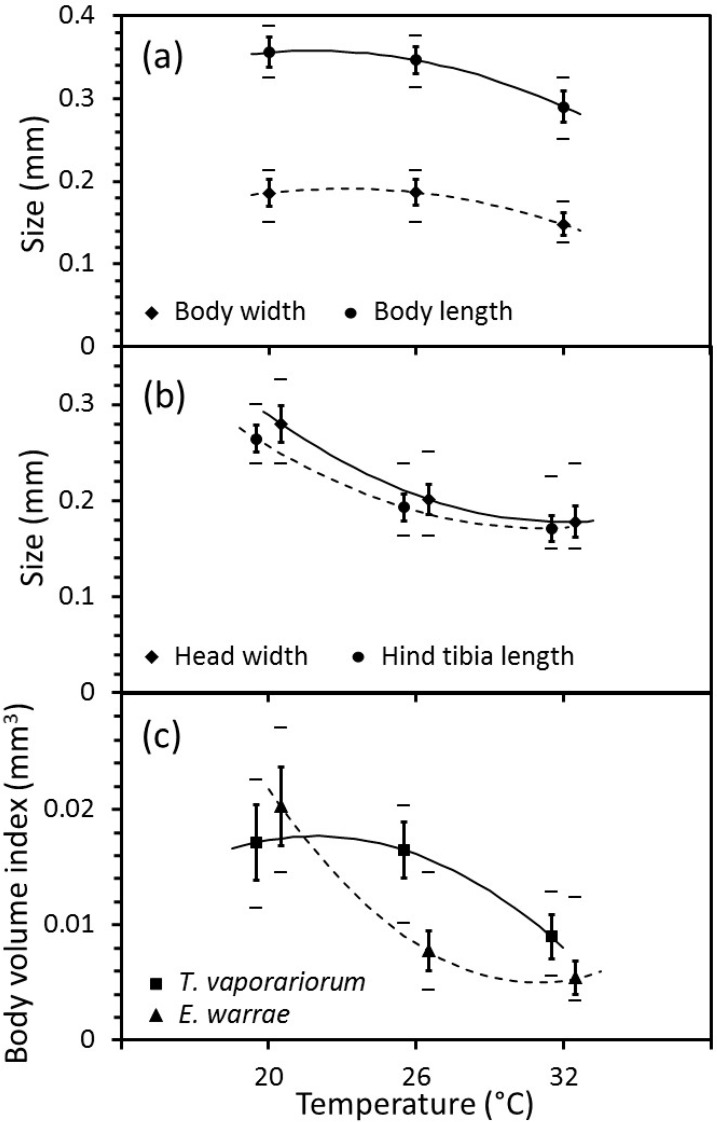
Effects of temperature on the body-sizes of *Trialeurodes vaporariorum* and *Eretmocerus warrae*. (**a**) Length and width of second instar *T. vaporariorum*. (**b**) Head width and hind-tibia length of adult *E. warrae*. (**c**) Body-volume indices of *T. vaporariorum* and *E. warrae*. Values are mean ± standard deviation (standard errors are too small to show on picture). Values of *E. warrae* are offset by ±0.5 °C in (**b**,**c**) to assist visualisation. Quadratic regression lines and regression models are shown (*p* < 0.001).

**Figure 3 insects-11-00039-f003:**
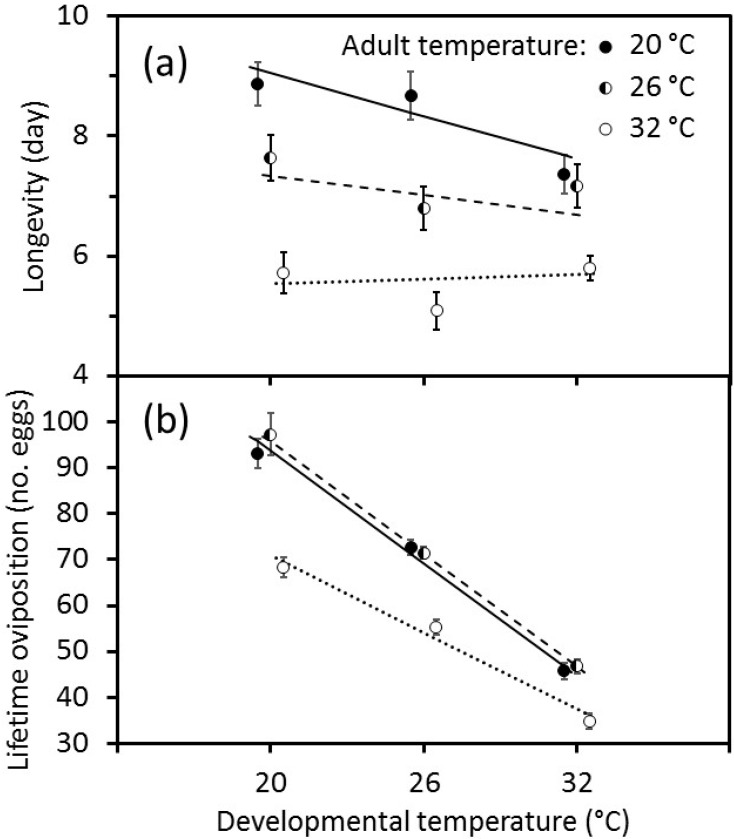
Effects of temperature during immature development on the (**a**) longevity and (**b**) lifetime oviposition of *Eretmocerus warrae*. Values are mean ± standard error. Longevity values are offset by ±0.5 °C to assist visualisation (*p* < 0.001).

**Figure 4 insects-11-00039-f004:**
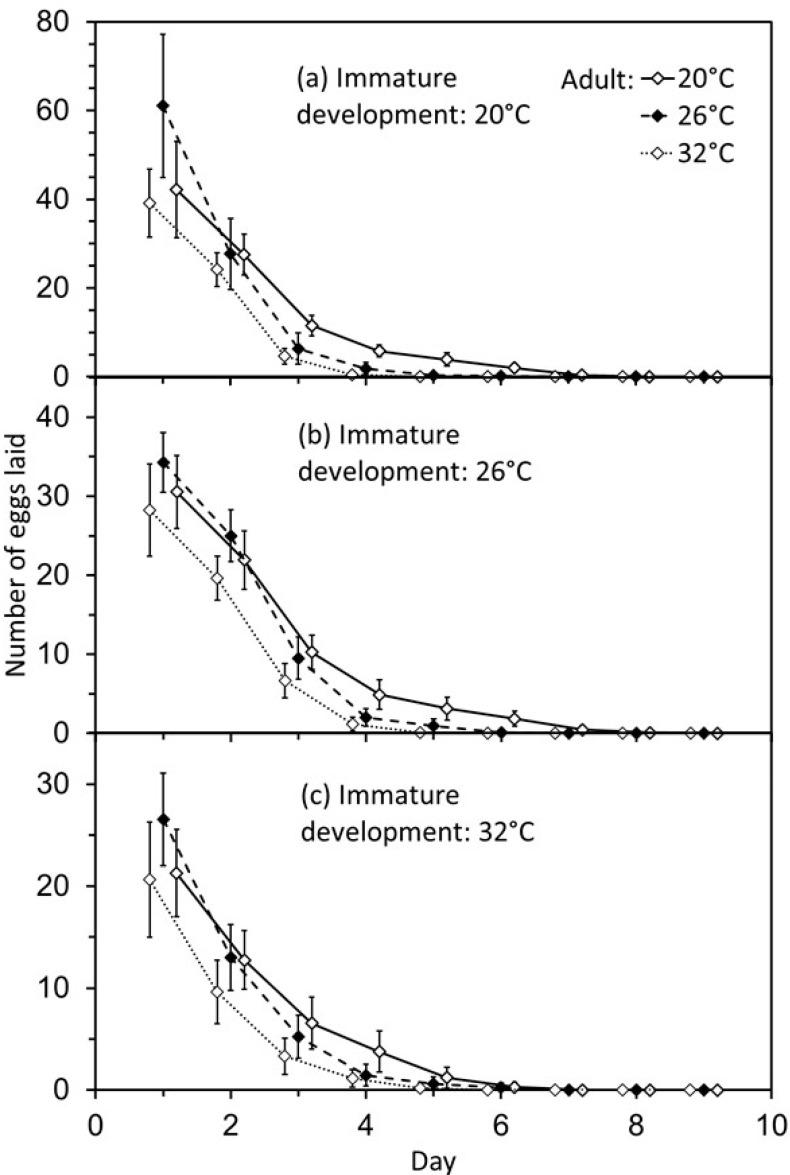
Effects of temperature during immature development and the adult stage on the daily oviposition of *Eretmocerus warrae*. (**a**) Immature development at 20 °C. (**b**) Immature development at 26 °C. (**c**) Immature development at 32 °C. Values are mean + standard deviation. Values are offset by ±0.15 days to assist visualisation (*p* < 0.001).

**Figure 5 insects-11-00039-f005:**
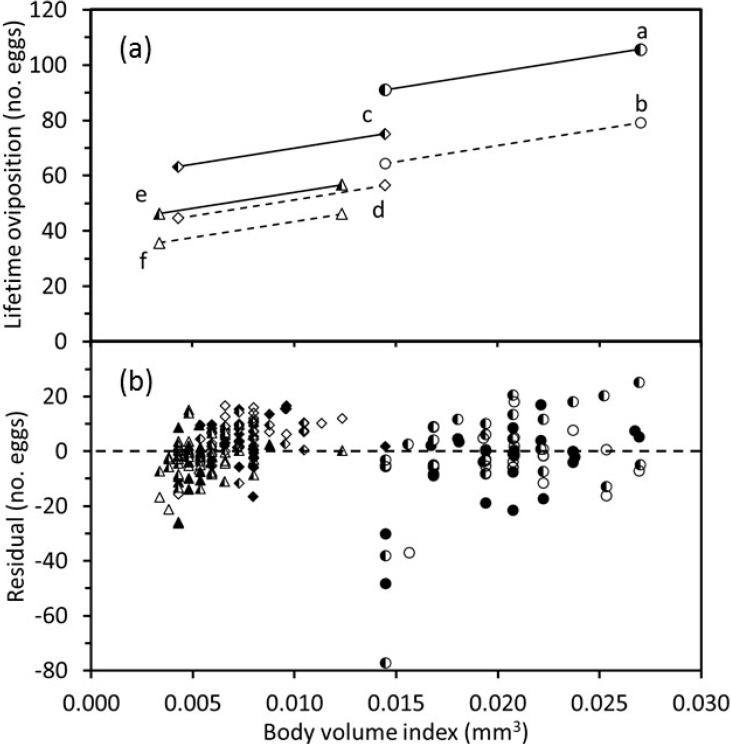
Relationship between the body-volume index and lifetime oviposition of *Eretmocerus warrae* as influenced by temperature during the immature development and adult stages. (**a**) Fitted regression lines projected over the ranges of observed body-volume indices. Treatments shown by each line (immature temperature:adult temperature in °C) are: a, 20:20 and 20:26; b, 20:32; c, 26:20 and 26:26; d, 26:32; e, 32:20 and 32:26; and f, 32:32. (**b**) Residuals from the regression analysis. Symbols indicate immature temperatures: 20—〇, 26—◇, 32—△; and adult temperatures: 20—filled symbols, 26—half-filled symbols, 32—open symbols (*p* < 0.001). Lines a, c, and e represent two adult temperatures as differences were not statistically significant.

**Figure 6 insects-11-00039-f006:**
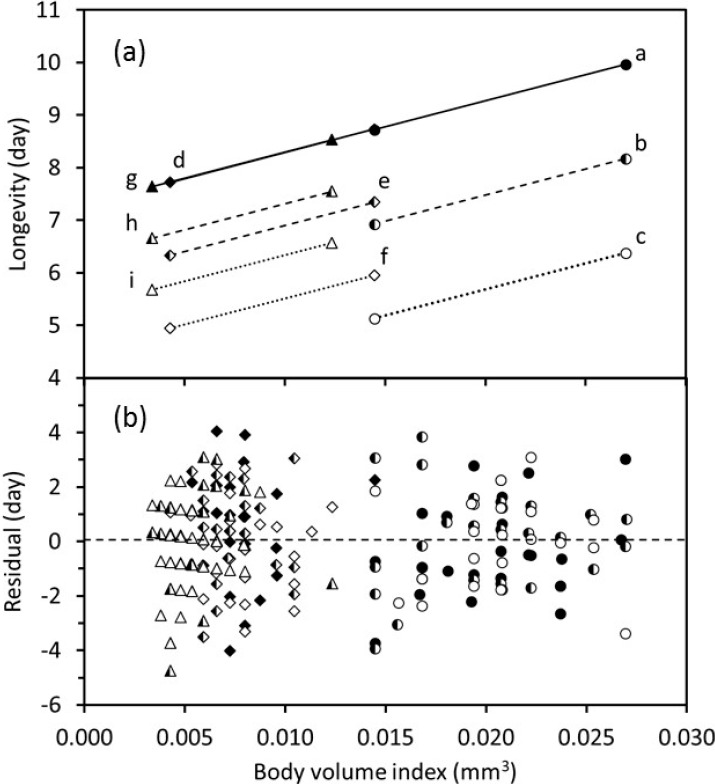
Relationship between body-volume index and adult longevity when *Eretmocerus warrae* developed at three constant temperatures and adults were subsequently held at one of three constant temperatures. (**a**) Fitted regression lines projected over the ranges of observed body-volume indices. Treatments shown by each line (immature temperature:adult temperature in °C) are: a, 20:20; b, 20:26; c, 20:32; d, 26:20; e, 26:26; f, 26:32; g, 32:20; h, 32:26; and i, 32:32. (**b**) Residuals from the regression analysis. Symbols indicate immature temperatures: 20—〇, 26—◇, 32—△; and adult temperatures: 20—filled symbols, 26—half-filled symbols, 32—open symbols (*p* < 0.001).

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
