# Peer review of "Larger is Better in the Parasitoid Eretmocerus warrae (Hymenoptera: Aphelinidae)"

_insects, 2020, doi:10.3390/insects11010039_

Round 1

Reviewer 1 Report

This study is soundly designed and offers practical insight for rearing parasitoid wasps for insect control in greenhouse settings. It is recommended for publication after several minor changes are made as described below.

In the methods section, the authors do not state whether they only selected female wasps that emerged (line 116) to then place in clip cages with the whitefly nymphs. This is assumed by the reader, but should be specified. 

Lines 92-96 are redundant with the previous paragraph.

Lines 129-130 in the methods section should be rewritten as it is unclear what "...treat the adult temperature" means.

Is it a standard procedure to multiply body volume indices by 1000? Provide a reference if possible.

In the results, it would be helpful to the reader if Rvalues were provided for each regression that is reported.

Figures 5 is confusing. The graph shows the range of body volumes. Are these the minimum and maximum values (and therefore based on a single insect for each value) or are they a certain percentile? Please describe. Also, it is not clear why lines have a different number of values depicted. For 5b, for example, it lists 20:32. This seems to mean that that line depicts the minimum and maximum body volume for insects that were raised at an immature temperature of 20 and an adult temperature of 32. What does 5a mean, where it is depicted as 20:20 & 26? How are 20 and 26 combined? Same for 5c and 5e. Please describe in greater detail.

The discussion needs to be rewritten to correct minor grammatical or phrasing errors. For example, line 201, change "quiet" to "quite". Also correct "grew" to "grown" or "reared". It also may be helpful to first summarize the overall results obtained in the current study. Then after that, compare these results with those obtained in other studies with other species.

Line 245, remove "my" and rephrase. 

Author Response

This study is soundly designed and offers practical insight for rearing parasitoid wasps for insect control in greenhouse settings. It is recommended for publication after several minor changes are made as described below.

In the methods section, the authors do not state whether they only selected female wasps that emerged (line 116) to then place in clip cages with the whitefly nymphs. This is assumed by the reader, but should be specified.Revised to clarify

Lines 92-96 are redundant with the previous paragraph.  Redundant text deleted

Lines 129-130 in the methods section should be rewritten as it is unclear what "...treat the adult temperature" means.  Revised

Is it a standard procedure to multiply body volume indices by 1000? Provide a reference if possible.  This advice was given by a statistician who isn’t available during the holiday period to suggest a reference.  As it is a linear scaling, it will not affect the statistical analysis.

In the results, it would be helpful to the reader if R2 values were provided for each regression that is reported.  Our aim is not to fit predictive models.  Rather we aim to identify relationships. We believe R2 values would be misleading.

Figures 5 is confusing. The graph shows the range of body volumes. Are these the minimum and maximum values (and therefore based on a single insect for each value) or are they a certain percentile? Please describe. Also, it is not clear why lines have a different number of values depicted. For 5b, for example, it lists 20:32. This seems to mean that that line depicts the minimum and maximum body volume for insects that were raised at an immature temperature of 20 and an adult temperature of 32. What does 5a mean, where it is depicted as 20:20 & 26? How are 20 and 26 combined? Same for 5c and 5e. Please describe in greater detail. Figure legend revised to clarify.

The discussion needs to be rewritten to correct minor grammatical or phrasing errors. For example, line 201, change "quiet" to "quite". Also correct "grew" to "grown" or "reared". It also may be helpful to first summarize the overall results obtained in the current study. Then after that, compare these results with those obtained in other studies with other species.  Discussion revised following recommendation.

Line 245, remove "my" and rephrase.  Text deleted.

Reviewer 2 Report

The paper deals with the influence of temperature on the size of a parasitoid wasps and its primary hosts, whitheflies in order to give useful information for scientists and breeders. I think that the paper is quite interesting but lack of discussion and the conclusions are not so appropriate. Measuring certain morphological characters such as lenght of tibiae as  weel width of the body in whiteflies is not easy and we have to consider that there is a huge variability within the species without temperature exposure. So I am not so well convinced about the utility of this kind of experiments, anyway after a critical revison of certain sentences and shifting the discussion towards  the insects basic knowledge could be important for the paper improvement.

I attach a file together with my considerations

Author Response

Line 41: s a citation necessary for introducing your paper? if yes, try to find another source because this is a species very different from those you considered, otherwise remove this citation. Text revised

Lines 48-50: who wrote this sentence? Please cite the AA!!! If you have wrote this sentence without any literature support, erase this sentence, please.... Reference cited

Line 81: E. warrae prefers to parasitise the 2nd instars of greenhouse whitefly. Please, insert citation. Citation added

Line 99: why did you chose these temperatures? Explanation provided with reference

Lines 113-115: is not so clear. Moreover I think that 4 adults (I suppose two males and 2 females, or 4 females?) could affect the oviposition behavior compared to the remaining groups. Text revised.  Oviposition behaviour isn’t relevant here as the aim was simply to achieve equivalent levels of parasitism in each treatment.

Line 123: I never heared the term "immature temperature" as well "adult temperature". What's the meaning? I think that it could be better to erase. We define the terminology here to use it as shorthand later.  Quotations added to clarify.

Discussion revised so the following specific comments are not addressed

Lines 144 - : what's the upper thermal threshold for this species? It could be better to take in consideration also this parameter in order to improve the also the discussion

Lines 162-164: very strange!! try to speculate why

Line 204: not so clear sentence

Lines 230 onwards: this is a very not useful reccomendation; first in the light of high costs required to do that, moreover producers have to control the parasitoid fitness by assessing the behavior as well some physiological parameters which are necessary in order to obtain a stable strain.

the discussion is too poor, you have to take in consideration also other whitefly parasitoids as well different kind of release in order to discuss the benefits that your reserch should give in practice.

Line 243: If I read well there are two Authors, right?

Lines 246 onwards : I found these sentences not useful neither for the scientist nor for producers. You have to rewrite the conclusion in order to improve the meaning of your achieved results as a basic research.

Reviewer 3 Report

The work is original and contribute to the biological control of the greenhouse whitefly using a parasitoid wasp Eretmocerus warrae. 

It is very well written and clear, well researched, and clearly illustrated. The topic of article fits to the journal scope. I found some minor errors.

L15: T. vaporariorum → italic

L41: add “(Hymenoptera: Megachilidae)

LL48-49: Please add reference(s)

L62: Delete “(Nauman & Schmidt)” 

L243, 245, 251: my → our?

L283: Trialeurodes vaporariorum → italic

Figs 2, 3, 5 and 6: Please indicate (a), (b) and (c) in each figures.

Fig 4: Temperature (℃) (the horizontal axis) → Days ?

Author Response

All of the following corrections were made

L15: T. vaporariorum → italic

L41: add “(Hymenoptera: Megachilidae)

LL48-49: Please add reference(s)

L62: Delete “(Nauman & Schmidt)”

L243, 245, 251: my → our?

L283: Trialeurodes vaporariorum → italic

Figs 2, 3, 5 and 6: Please indicate (a), (b) and (c) in each figures.

Fig 4: Temperature (℃) (the horizontal axis) → Days ?

Round 2

Reviewer 2 Report

I have no comments for the AA; they have followed at least all my suggestions and now the paper is ready to be published

Author Response

I suggest only very minor changes. In the Abstract, please add the aims of the reported research (something like "... we investigated how temperature affects the body-size, life-time oviposition and longevity of.... ) and the final importance of the obtained results. In the present form the Abstract seems just a list of features of the biology of both the parasitoid and its host. Check also the sentence "The life-time oviposition of female adult of E. warrae that grew at immature developmental temperature of 20 °C was 86 ± 22 eggs, more than that 66 ± 11 eggs at 26 °C and 65 ± 23 eggs at 32 °C". Did you mean "...more eggs than that at 26°C (66 ± 11) and at  32 °C (65 ± 23)"?  

All the minor changes have been made.